# Repeated social defeat stress enhances glutamatergic synaptic plasticity in the VTA and cocaine place conditioning

**Claire E Stelly[1,2], Matthew B Pomrenze[2,3], Jason B Cook[1,2], Hitoshi Morikawa[1,2]\***

[1]Department of Neuroscience, University of Texas, Austin, United States; [2]Waggoner Center for Alcohol and Addiction Research, University of Texas, Austin, United States; [3]Division of Pharmacology and Toxicology, University of Texas, Austin, United States

**Abstract** Enduring memories of sensory cues associated with drug intake drive addiction. It is well known that stressful experiences increase addiction vulnerability. However, it is not clear how repeated stress promotes learning of cue-drug associations, as repeated stress generally impairs learning and memory processes unrelated to stressful experiences. Here, we show that repeated social defeat stress in rats causes persistent enhancement of long-term potentiation (LTP) of NMDA receptor-mediated glutamatergic transmission in the ventral tegmental area (VTA). Protein kinase A-dependent increase in the potency of inositol 1,4,5-triphosphate-induced $Ca^{2+}$ signaling underlies LTP facilitation. Notably, defeated rats display enhanced learning of contextual cues paired with cocaine experience assessed using a conditioned place preference (CPP) paradigm. Enhancement of LTP in the VTA and cocaine CPP in behaving rats both require glucocorticoid receptor activation during defeat episodes. These findings suggest that enhanced glutamatergic plasticity in the VTA may contribute, at least partially, to increased addiction vulnerability following repeated stressful experiences.

**\*For correspondence:** morikawa@utexas.edu

**Competing interests:** The authors declare that no competing interests exist.

## Introduction

Humans with a history of stressful or traumatic experiences are more prone to develop substance use disorders (*Sinha, 2008*). Adverse experience recruits the hypothalamic-pituitary-adrenal (HPA) axis stress response, culminating in release of glucocorticoids that enables the body to cope with insults to homeostasis (*Munck et al., 1984*). In rodent models, repeated activation of the stress response typically disrupts learning and cognition [e.g., spatial learning (*Conrad et al., 1996*), working memory (*Mizoguchi et al., 2000*), and cognitive flexibility (*Liston et al., 2006*)]. In contrast to these deficits, prior stress enhances the learning of Pavlovian cue-outcome associations driven by rewarding stimuli, assessed with conditioned place preference (CPP) (*Kreibich et al., 2009*; *Burke et al., 2011*; *Chuang et al., 2011*), or aversive/stressful stimuli, assessed with fear conditioning (*Conrad et al., 1999*; *Sandi et al., 2001*; *Suvrathan et al., 2014*). These effects of stress may have arisen from evolutionary pressure to rapidly acquire information predicting food, shelter, and predator threat during periods of duress. Augmented Pavlovian reward learning mechanisms in stressed individuals may also heighten susceptibility to addiction, as acquisition of cue-drug associations is a crucial early step in drug use, and powerful, enduring memories of drug-associated cues trigger craving and relapse as recreational use progresses to addiction (*Hyman et al., 2006*). However, it is not clear how repeated stressful experience promotes the learning of cue-drug/reward associations, as repeated stress is generally detrimental to synaptic plasticity underlying learning and

**eLife digest** Daily stress increases the likelihood that people who take drugs will become addicted. A very early step in the development of addiction is learning that certain people, places, or paraphernalia are associated with obtaining drugs. These 'cues' – drug dealers, bars, cigarette advertisements, etc. – become powerful motivators to seek out drugs and can trigger relapse in recovering addicts. It is thought that learning happens when synapses (the connections between neurons in the brain) that relay information about particular cues become stronger. However, it is not clear how stress promotes the learning of cue-drug associations.

Stelly et al. investigated whether repeated episodes of stress make it easier to strengthen synapses on dopamine neurons, which are involved in processing rewards and addiction. For the experiments, rats were repeatedly exposed to a stressful situation – an encounter with an unfamiliar aggressive rat – every day for five days. Stelly et al. found that these stressed rats formed stronger associations between the drug cocaine and the place where they were given the drug (the cue). Furthermore, a mechanism that strengthens synapses was more sensitive in the stressed rats than in unstressed rats. These changes persisted for 10-30 days after the stressful situation, suggesting that stress might begin a period of time during which the individual is more vulnerable to addiction.

The experiments also show that a hormone called corticosterone – which is released during stressful experiences – is necessary for stress to trigger the changes in the synapses and behavior of the rats. However, corticosterone must work with other factors because giving this hormone to unstressed rats was not sufficient to trigger the changes seen in the stressed rats. Future experiments will investigate what these other stress factors are and how they work together with corticosterone.

memory unrelated to stressful events (*Kim and Diamond, 2002*; *Joels et al., 2006*; *Schwabe et al., 2012*; *Chattarji et al., 2015*).

The mesolimbic dopamine system originating in the ventral tegmental area (VTA) is critical for reward processing. VTA dopamine neurons tonically fire action potentials (APs) at 1–5 Hz, while responding to unexpected rewards with phasic burst firing (2–10 APs at 10–50 Hz). These dopamine neuron responses are hypothesized to drive the learning of Pavlovian cue-reward associations (*Tsai et al., 2009*; *Darvas et al., 2014*). Intriguingly, over the course of repeated cue-reward pairing, dopamine neurons acquire a conditioned burst response to reward-predictive cues, which is thought to encode the positive motivational valence of those cues and to invigorate reward-seeking behavior (*Schultz, 1998*; *Berridge et al., 2009*; *Bromberg-Martin et al., 2010*).

Glutamatergic inputs activating NMDA receptors (NMDARs) drive the transition from tonic firing to bursting in dopamine neurons (*Overton and Clark, 1997*; *Zweifel et al., 2009*; *Wang et al., 2011*); therefore, potentiation of cue-driven NMDAR inputs may contribute to the acquisition of conditioned bursting. Indeed, NMDAR-mediated transmission undergoes long-term potentiation (LTP) when cue-like glutamatergic input stimulation is repeatedly paired with reward-like bursting in dopamine neurons (*Harnett et al., 2009*). LTP induction requires amplification of burst-evoked $Ca^{2+}$ signals by preceding activation of group I metabotropic glutamate receptors (mGluRs; more specifically mGluR1) coupled to the generation of inositol 1,4,5-triphosphate ($IP_3$). Here, $IP_3$ receptors ($IP_3$Rs) detect the coincidence of $IP_3$ generated by glutamatergic input activating mGluRs and burst-driven $Ca^{2+}$ entry. $IP_3$ enhances $Ca^{2+}$-induced activation of $IP_3$Rs by promoting access to the stimulatory $Ca^{2+}$ sites, thereby promoting $Ca^{2+}$-induced $Ca^{2+}$ release from intracellular stores (*Taylor and Laude, 2002*). In this study, we demonstrate that repeated social defeat stress (1) enhances NMDAR LTP in the VTA via an increase in $IP_3$ sensitivity of $IP_3$Rs and (2) promotes acquisition of cocaine CPP in behaving rats, and both of these effects require glucocorticoid action during defeat stress.

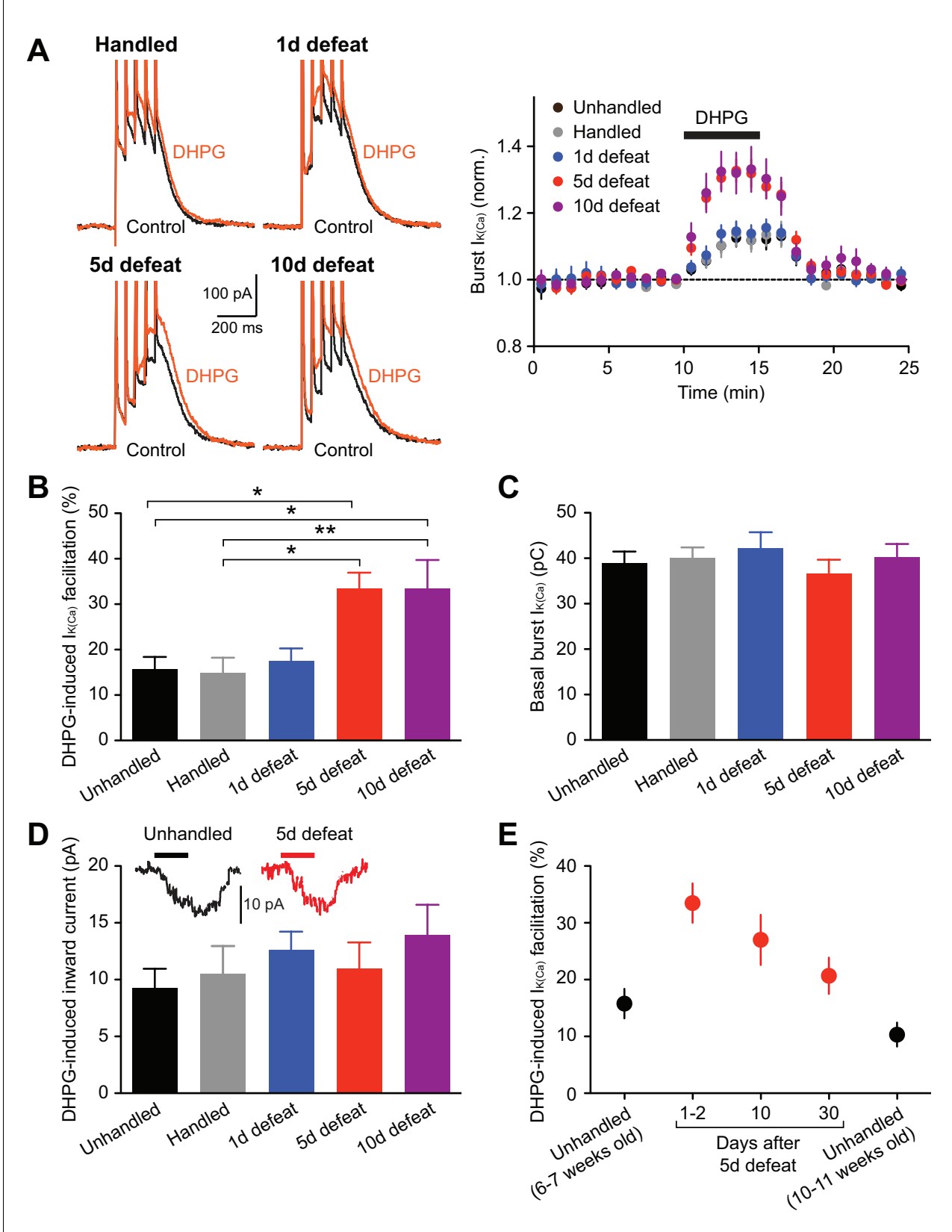

**Figure 1.** mGluR-dependent facilitation of burst-evoked Ca$^{2+}$ signals is enhanced after repeated social defeat. (**A**) Example traces (left) and summary time graph (right) illustrating the facilitating effect of DHPG (1 μM) on burst I$_{K(Ca)}$ in neurons from unhandled rats (traces not shown), rats handled for 5

*Figure 1 continued on next page*

*Figure 1 continued*

days, and rats that underwent social defeat for 1, 5, or 10 days. (B) Summary bar graph showing the magnitude of DHPG-induced burst $I_{K(Ca)}$ facilitation (unhandled: 20 cells from 12 rats, handled: 20 cells from 13 rats, 1 day defeat: 19 cells from 11 rats, 5 day defeat: 21 cells from 13 rats, 10 day defeat: 19 cells from 10 rats; $F_{4,94} = 6.19$, $p<0.001$, one-way ANOVA). *$p<0.05$, **$p<0.01$ (Bonferroni post hoc test). (C) The size of basal burst $I_{K(Ca)}$ was not altered by social defeat. (D) DHPG-induced inward currents were not affected by social defeat. Inset: Example traces of DHPG-induced currents from unhandled and 5-day defeated rats (5-min DHPG perfusion at the horizontal bar). (E) Summary graph depicting DHPG effect on burst $I_{K(Ca)}$ after different intervals following 5-day social defeat. Data in 6–7 weeks old unhandled and 1–2 day interval groups were from those in panels A–D (6–7 weeks old unhandled: 20 cells from 12 rats, 1–2 day interval: 21 cells from 13 rats, 10-day interval: 17 cells from 8 rats, 30-day interval: 16 cells from 9 rats, 10–11 weeks old unhandled: 15 cells from 7 rats).

## Results

### Repeated social stress increases mGluR-dependent facilitation of burst-evoked Ca²⁺ signals

NMDAR LTP induction requires mGluR/$IP_3$-induced facilitation of burst-evoked $Ca^{2+}$ signals (*Harnett et al., 2009*). Therefore, we first examined the effect of the group I mGluR agonist DHPG (1 µM; 5-min perfusion) on burst-evoked $Ca^{2+}$ signals, assessed by the size of $Ca^{2+}$-activated SK currents (termed burst $I_{K(Ca)}$) in control and stressed animals. Rats were unhandled, handled, or socially defeated (at the end of the dark cycle) for 1, 5, or 10 consecutive days, and VTA slices were prepared 1–2 days after the final handling/defeat session. The magnitude of DHPG effect on $I_{K(Ca)}$ was significantly larger in animals that underwent 5 or 10 days of defeat stress compared to unhandled and handled controls, whereas a single defeat session failed to alter the DHPG effect (*Figures 1A and B*). There was no significant difference between unhandled and handled controls. The effect of stress plateaued by 5 days, as comparable enhancement of DHPG effect was observed after 10-day defeat. Basal burst $I_{K(Ca)}$ was consistent across groups (*Figure 1C*), suggesting no alterations in AP-evoked $Ca^{2+}$ influx. DHPG-induced inward currents, which are independent of $Ca^{2+}$ signaling (*Guatteo et al., 1999*), were not affected (*Figure 1D*); thus the stress-induced increase in $I_{K(Ca)}$ facilitation results from changes in $IP_3$ signaling downstream of mGluRs.

Next, to examine the persistence of repeated stress effect, the interval between the last social defeat session and recording was prolonged to 10 and 30 days. Although stress-induced enhancement displayed gradual recovery, DHPG effect was still elevated after 30 days compared to age-matched controls (*Figure 1E*). Subsequent electrophysiology experiments were performed in 5-day defeated rats (with 1–2 day interval) and controls (unhandled and handled controls combined).

### Protein kinase a mediates IP₃R sensitization in socially defeated animals

To directly examine alterations in $IP_3$ signaling, we applied different concentrations of $IP_3$ (expressed in µM·µJ; see Methods and materials) into the cytosol using flash photolysis of caged $IP_3$, and $IP_3$R-mediated $Ca^{2+}$ release was assessed by flash-evoked SK currents($I_{IP3}$) (*Figure 2A*). The average $IP_3$ concentration-response curve displayed a leftward shift in defeated rats compared to controls (*Figure 2B*). Accordingly, the average $EC_{50}$ valuewas significantly smaller in the defeated group (*Figure 2C*). Maximal $I_{IP3}$ amplitude did not differ between groups (*Figure 2D*), indicating a change in the potency, but not the efficacy, of $IP_3$ in eliciting $Ca^{2+}$ release.

Protein kinase A (PKA)-dependent phosphorylation of $IP_3$Rs increases their $IP_3$ sensitivity (*Wagner et al., 2008*). To determine the involvement of PKA in stress-induced $IP_3$R sensitization, the effect of a selective peptide inhibitor of PKA, PKI-(6–22)-amide (PKI; 200 µM, loaded into the cytosol via the whole-cell pipette for >15–20 min after break-in), was tested. PKI reversed the leftward shift in $IP_3$ concentration-response curve, and thus the decrease in $EC_{50}$ value, in stressed animals, while having no significant effect on maximal $I_{IP3}$ amplitude (*Figures 2B–D*). PKI had no effect in the control group, suggesting low basal PKA activity in non-stressed animals. It should be noted that PKI eliminated the difference in $IP_3$ potency between the two groups. These data indicate that repeated social stress sensitizes $IP_3$Rs via a PKA-dependent mechanism.

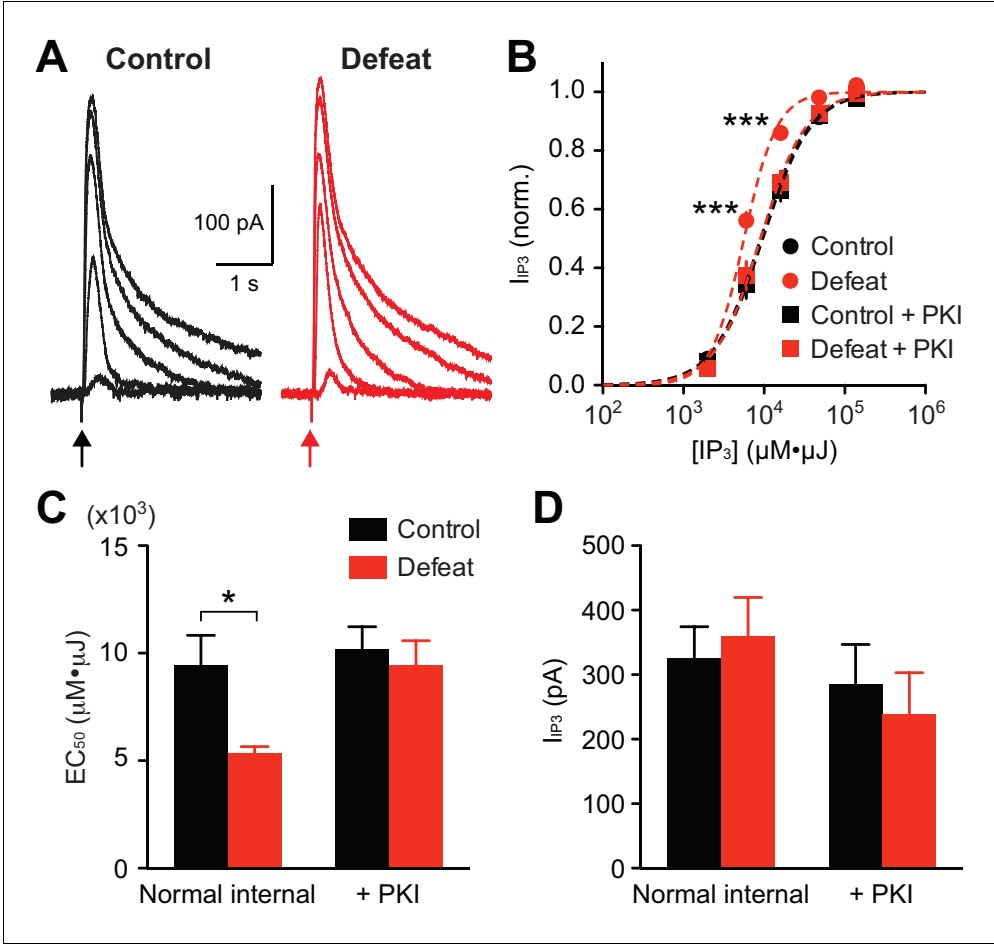

**Figure 2.** PKA activity maintains increased IP$_3$R sensitivity in socially defeated rats. (**A**) Example traces of I$_{IP3}$ evoked by different concentrations of IP$_3$ (2000, 6000, 16000, 48000, and 140000 µM·µJ) in control and defeated rats. (**B**) Averaged IP$_3$ concentration-response curves from control and defeated rats. I$_{IP3}$ amplitudes were normalized to the maximal value (estimated from fit to a logistic equation) in each cell. Recordings were made with normal internal solution or with PKI (control: 12 cells from 8 rats, defeat: 12 cells from 7 rats, control + PKI: 15 cells from 9 rats, defeat + PKI: 14 cells from 8 rats; treatment (defeat/PKI): $F_{3,196}$ = 4.88, $p<0.01$; IP$_3$ concentration: $F_{4,196}$ = 1214, $p<0.001$; treatment × IP$_3$ concentration: $F_{12,196}$ = 4.42, $p<0.001$; mixed two-way ANOVA). ***$p<0.001$ vs. control (Bonferroni post hoc test). Lines represent logistic fit to the averaged data in each group. (**C**) Summary bar graph depicting the average EC$_{50}$ values (EC$_{50}$ determined in each cell) in the 4 groups shown in (**B**) (defeat: $F_{1,49}$ = 5.11, $p<0.05$; PKI: $F_{1,49}$ = 5.11, $p<0.05$; two-way ANOVA). *$p<0.05$ (Bonferroni post hoc test). (**D**) The maximal I$_{IP3}$ amplitude was not affected by social defeat experience or PKI during recording.

## Repeated social stress enhances NMDAR LTP

We next examined whether repeated social defeat affects NMDAR LTP induction, which requires mGluR/IP$_3$-dependent facilitation of burst-evoked Ca$^{2+}$ signals and is gated by PKA (*Harnett et al., 2009*). Application of a low concentration of IP$_3$ preceding APs can effectively facilitate I$_{K(Ca)}$ (*Cui et al., 2007*; *Ahn et al., 2010*; *Bernier et al., 2011*). Thus, the LTP induction protocol consisted of applying a low concentration of IP$_3$ (250 µM·µJ) 50 ms prior to simultaneous pairing of a burst with a brief train of synaptic stimuli (*Figure 3A*), the latter being necessary to activate NMDARs at stimulated synapses at the time of burst for LTP induction (*Harnett et al., 2009*; *Whitaker et al., 2013*). This induction protocol produced little LTP in control animals, while large LTP was induced in defeated animals (*Figure 3B and C*). IP$_3$ application, which caused little I$_{IP3}$ by itself, caused facilitation of burst I$_{K(Ca)}$ (assessed immediately before LTP induction), which was significantly larger in cells from defeated animals (*Figure 3D*). Furthermore, the magnitude of LTP was positively correlated with that of IP$_3$-induced facilitation of I$_{K(Ca)}$ across neurons from both groups (*Figure 3E*). Robust

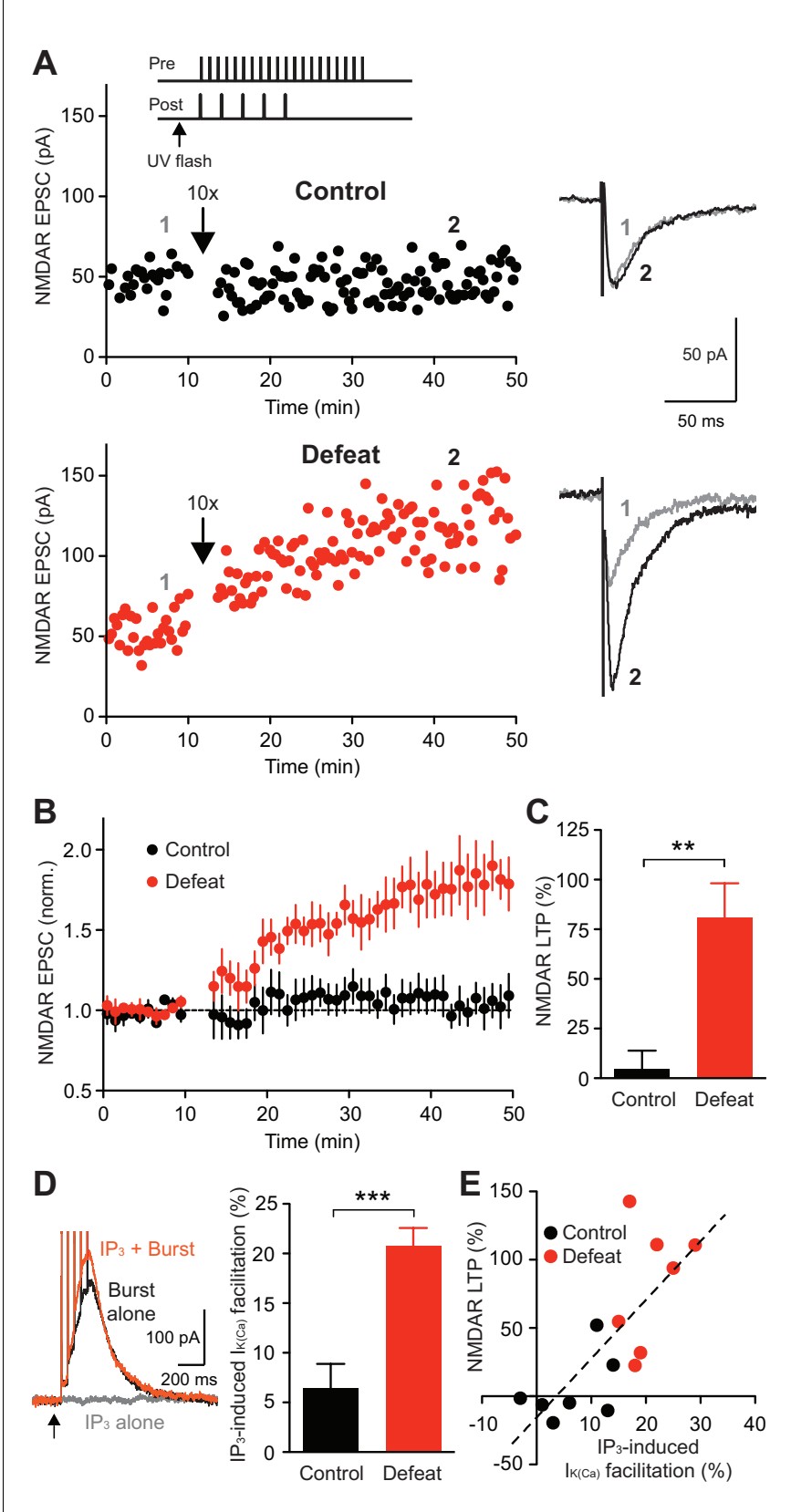

**Figure 3.** NMDAR-mediated transmission is more susceptible to LTP induction after social defeat. (**A**) Example experiments to induce NMDAR LTP in neurons from control and defeated rats. Time graphs of NMDAR EPSCs are
*Figure 3 continued on next page*

*Figure 3 continued*

shown with example traces at times indicated by numbers (baseline: gray, post-induction: black). The LTP induction protocol (IP$_3$-synaptic stimulation-burst combination; illustrated in top inset) was delivered after 10-min baseline recording (at arrow). (**B**) Summary time graph of baseline-normalized NMDAR EPSCs in LTP experiments (control: 7 cells from 7 rats, defeat: 7 cells from 7 rats). (**C**) Summary of NMDAR LTP magnitude in control and defeated rats ($t_{12}$ = 3.93, **p<0.01, unpaired t-test). (**D**) Example traces (left; from the defeated rat shown in **A**) and summary (right) of I$_{K(Ca)}$ facilitation by IP$_3$ assessed before LTP induction ($t_{12}$ = 4.65, ***p<0.001, unpaired t-test). (**E**) The magnitude of NMDAR LTP is plotted versus the magnitude of IP$_3$-induced facilitation of I$_{K(Ca)}$. Dashed line is a linear fit to all data points from both control and defeated rats.

The following figure supplement is available for figure 3:

**Figure supplement 1.** Summary time graph of baseline-normalized NMDAR EPSCs in LTP experiments using a high concentration of IP$_3$ (500 μM·μJ) during induction in control rats (4 cells from 4 rats).

LTP was induced in control rats when a higher IP$_3$ concentration (500 μM·μJ), which produced larger I$_{K(Ca)}$ facilitation, was used during induction (***Figure 3—figure supplement 1***). These results suggest that the enhanced LTP in defeated rats is a consequence of increased IP$_3$R sensitivity enabling greater facilitation of burst-evoked Ca$^{2+}$ signals.

It has been reported that repeated stress alters NMDAR expression in certain brain areas (***Fitzgerald et al., 1996***; ***Yuen et al., 2012***; ***Costa-Nunes et al., 2014***; ***Chattarji et al., 2015***). We found that bath application of NMDA (10 μM) produced comparable inward currents in control and defeated rats (***Figure 4***); thus repeated defeat stress caused no significant changes in global NMDAR-mediated excitation.

Repeated stress appears to differentially modulate tonic firing of VTA dopamine neurons recorded in vivo, as both an increase and decrease have been reported with different stress paradigms (***Cao et al., 2010***; ***Valenti et al., 2012***; ***Tye et al., 2013***). However, repeated social defeat failed to alter tonic firing measured in ex vivo slices (***Figure 5A***). Furthermore, the amplitude of hyperpolarization-activated cationic currents (I$_h$), which contribute to intrinsic dopamine neuron pacemaker activity (***Neuhoff et al., 2002***), was not affected (***Figure 5B***).

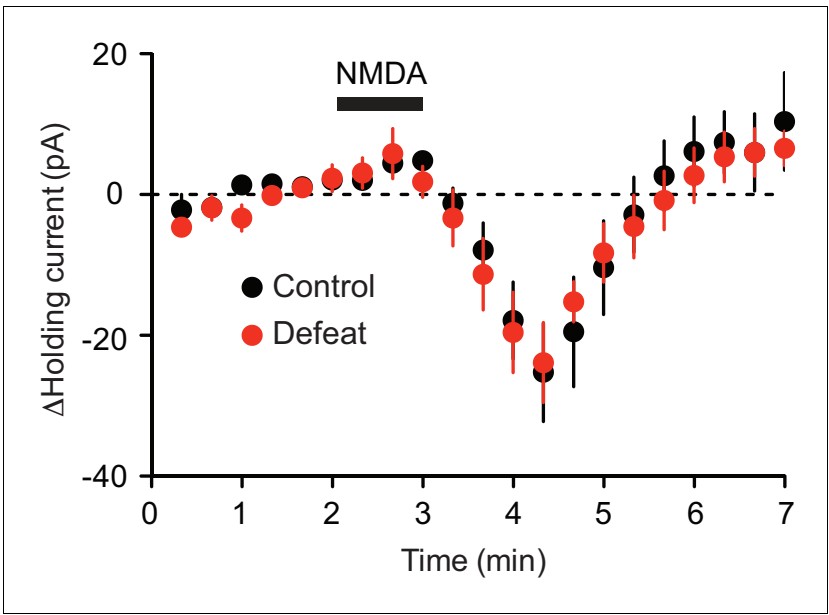

**Figure 4.** Summary time graph depicting inward currents induced by 1-min perfusion of NMDA (10 μM) in VTA dopamine neurons from control (8 cells from 3 rats) and 5 day defeated rats (6 cells from 2 rats).

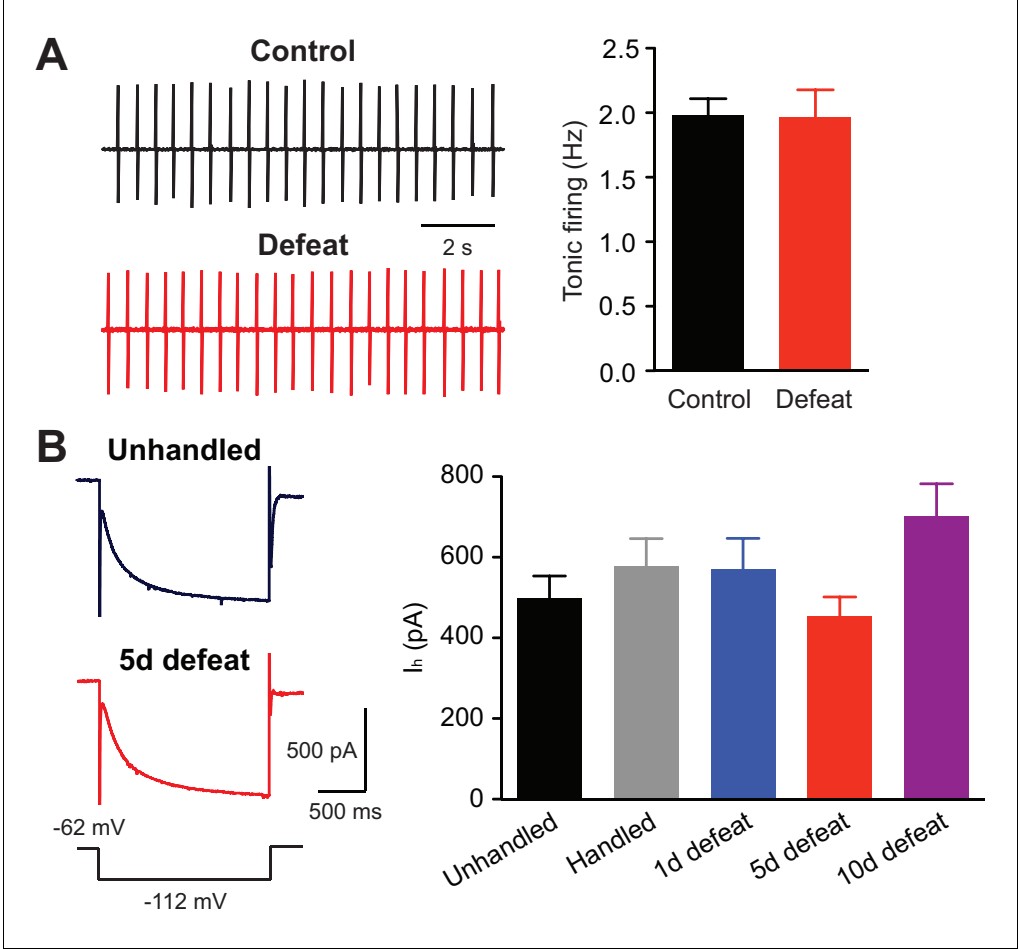

**Figure 5.** Tonic firing is unaltered by social defeat. (**A**) Example traces (left) and summary (right) of tonic firing frequency in VTA neurons from control and defeated rats (control: 15 cells from 3 rats, 5 defeats: 9 cells from 3 rats; $t_{22}$ = 0.066, p=0.95, unpaired t-test). In these experiments, loose-patch recordings (<20 MΩ seal) were made using pipettes filled with 150 mM NaCl to monitor tonic pacemaker firing. (**B**) Example traces (left; voltage step depicted at bottom) and summary (right) of $I_h$ currents recorded in cells from animals that underwent control procedures or 1, 5, or 10 days of defeat (data were obtained from the same cells shown in *Figures 1A–D;* $F_{4,94}$ = 2.01, p=0.10, one-way ANOVA).

## Glucocorticoid receptor activation is necessary but not sufficient for stress-induced IP₃R sensitization

A major consequence of stress-induced HPA axis activation is the secretion of glucocorticoids (corticosterone in rodents) into the blood (*Munck et al., 1984*). Thus, we sought to determine whether corticosterone, which readily crosses the blood-brain barrier, is involved in the increase in IP₃R sensitivity with repeated stress. Corticosterone activates both glucocorticoid receptors (GRs) and mineralocorticoid receptors (MRs); however, MRs are typically saturated by circadian fluctuations in corticosterone, while lower-affinity GRs are activated by levels attained with stress (*Joels and de Kloet, 1994*). Therefore, the role of GRs was examined by treating rats with the antagonist mifepristone (40 mg/kg, i.p.) or vehicle 30 min prior to each defeat session. The DHPG effect on burst $I_{K(Ca)}$ was significantly enhanced by repeated defeat in the vehicle-treated group, while there was no effect of stress in the mifepristone-treated group (*Figure 6A*). Thus, blockade of GRs during defeat sessions prevented IP₃R sensitization. Next, rats were treated with corticosterone (2.5, 5, or 15 mg/kg, i.p.; at the end of the dark cycle) for 5 days (no social defeat). The lowest dose (2.5 mg/kg) produces elevation in blood corticosterone concentration comparable to that evoked by a moderate stressor (*Graf et al., 2013*), while higher doses (≥10 mg/kg) have been used to simulate severe

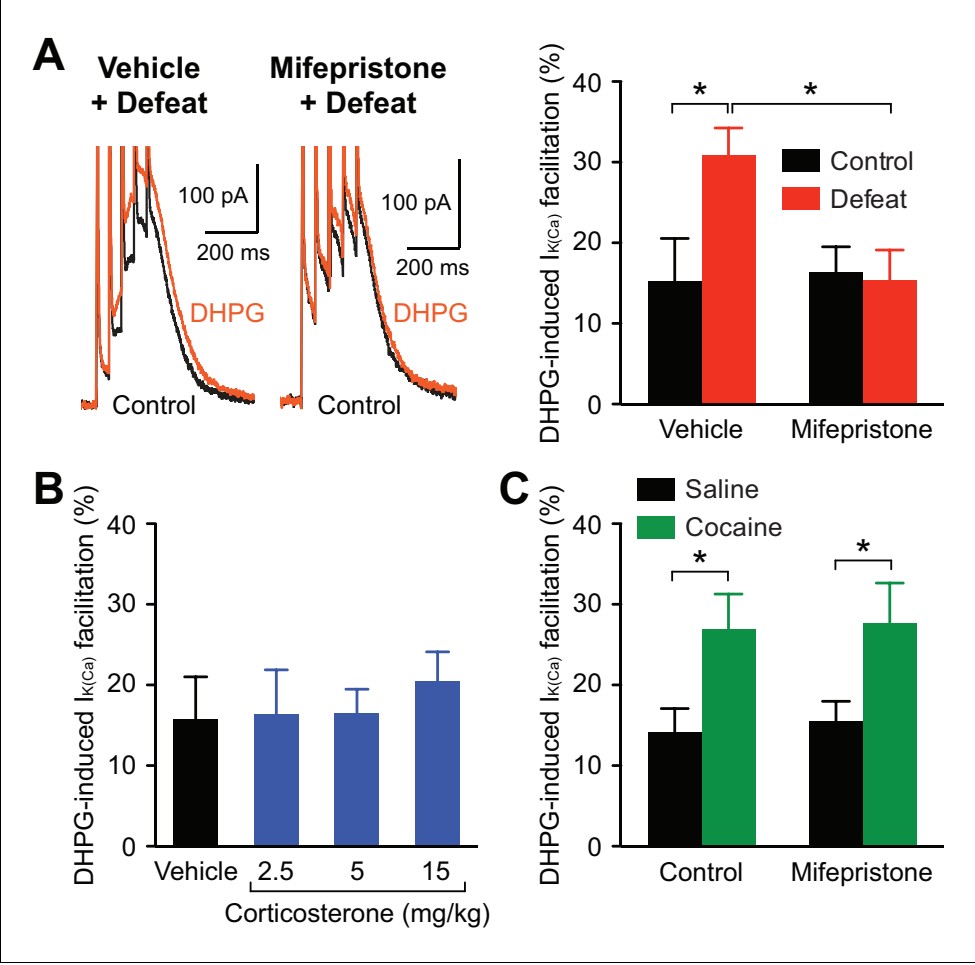

**Figure 6.** Stress-induced, but not cocaine-induced, IP$_3$R sensitization is prevented by GR blockade. (**A**) Example traces (left) and summary (right) of DHPG-induced burst I$_{K(Ca)}$ facilitation in neurons from animals that were injected with vehicle or mifepristone before undergoing control handling or social defeat sessions (vehicle + control: 8 cells from 5 rats, vehicle + defeat: 10 cells from 4 rats, mifepristone + control: 10 cells from 5 rats, mifepristone + defeat: 11 cells from 4 rats; defeat × mifepristone: F$_{1,35}$ = 4.56, p<0.05, two-way ANOVA). *p<0.05 (Bonferroni post hoc test). (**B**) Summary bar graph showing that repeated corticosterone treatment (once daily for 5 days) failed to affect DHPG-induced burst I$_{K(Ca)}$ facilitation (vehicle: 8 cells from 5 rats, 2.5 mg/kg: 11 cells from 5 rats, 5 mg/kg: 10 cells from 4 rats, 15 mg/kg: 12 cells from 7 rats). (**C**) Summary bar graph demonstrating that mifepristone pretreatment failed to block the increase in DHPG effect resulting from repeated cocaine treatment (10 mg/kg, i.p., once daily for 5 days) (saline: 17 cells from 7 rats, cocaine: 16 cells from 7 rats, mifepristone + saline: 21 cells from 8 rats, mifepristone + cocaine: 16 cells from 6 rats; cocaine: F$_{1,66}$ = 11.4, p<0.01, two-way ANOVA). *p<0.05 (Bonferroni post hoc test).

stress levels (*Akirav et al., 2004*). None of the tested doses significantly altered the DHPG effect on burst I$_{K(Ca)}$ (*Figure 6B*). Together, these results show that GR signaling is necessary, but not sufficient, for IP$_3$R sensitization.

It has been shown that repeated psychostimulant treatment sensitizes IP$_3$Rs and enhances NMDAR LTP (*Ahn et al., 2010*). As addictive drugs, including psychostimulants, stimulate corticosterone secretion (*Armario, 2010*), we next examined the role of GR signaling in psychostimulant-induced IP$_3$R sensitization. Rats were treated with mifepristone (40 mg/kg, i.p.) 30 min prior to injection of cocaine (10 mg/kg, i.p.) or saline for 5 days. The effect of DHPG on burst I$_{K(Ca)}$ was significantly larger in cocaine-treated animals, which was not affected by mifepristone pretreatment (*Figure 6C*). Thus, GR signaling is not involved in psychostimulant-induced IP$_3$R sensitization.

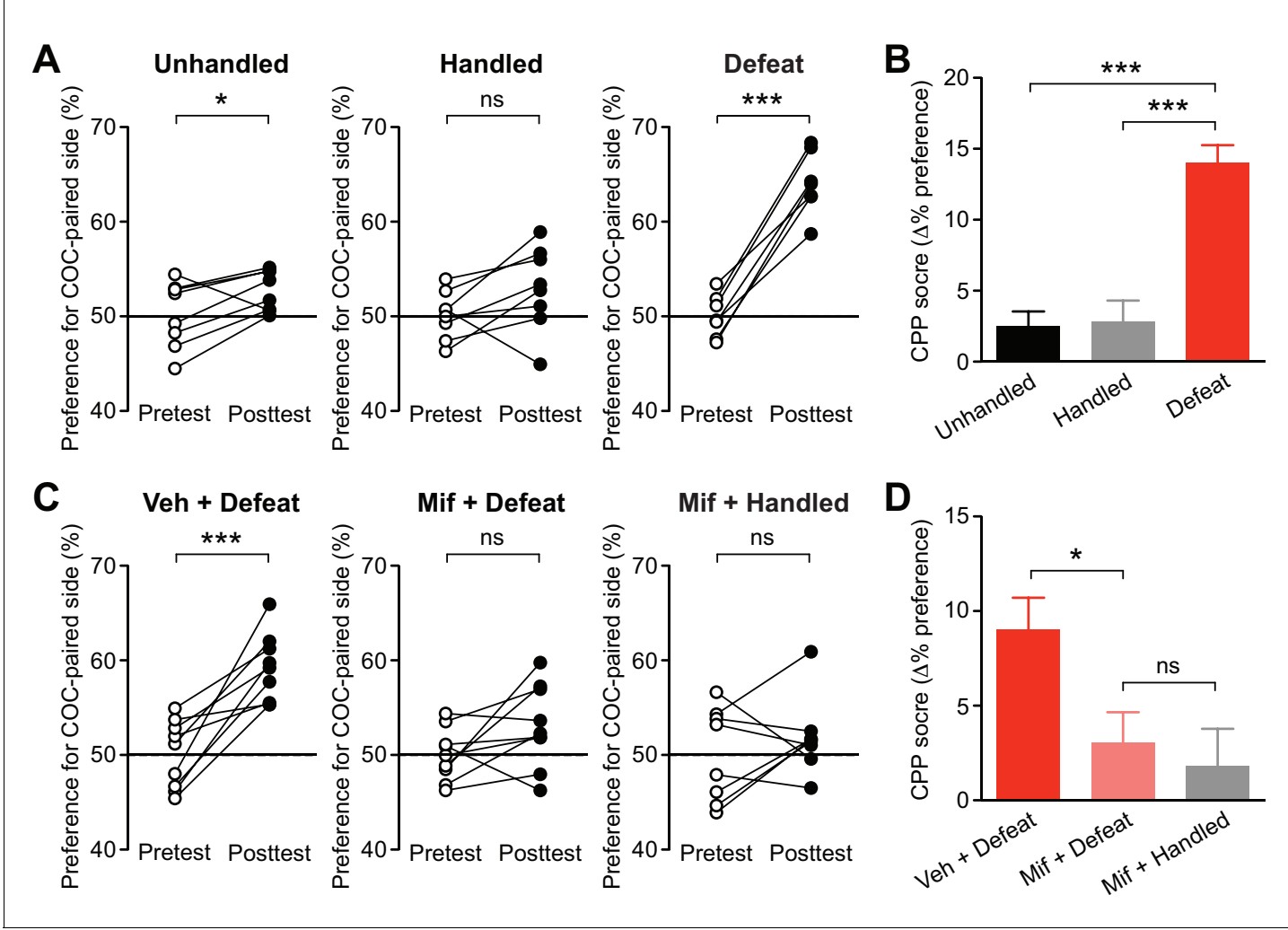

**Figure 7.** Social defeat promotes cocaine-induced CPP via a GR-dependent mechanism. (A) Summary of changes in the preference for the cocaine-paired side following 1-day conditioning in unhandled, handled, and defeated rats (unhandled: $t_7$ = 2.51, p<0.05; handled: $t_7$ = 1.90, p=0.10; defeat: $t_6$ = 11.0, p<0.001; paired t-test). (B) Summary of 1-day cocaine CPP scores in unhandled, handled, and defeated rats ($F_{2,20}$ = 25.2, p<0.001, one-way ANOVA). ***p<0.001 (Bonferroni post hoc test). (C) Summary of changes in the preference for the cocaine-paired side following 1-day conditioning in rats pretreated with vehicle or mifepristone 30 min prior to social defeat or handling sessions (vehicle + defeat: $t_8$ = 5.30, p<0.001; mifepristone + defeat: $t_8$ = 1.90, p=0.09; mifepristone + handled: $t_7$ = 0.95, p=0.37; paired t-test). (D) Summary of 1-day cocaine CPP scores in the 3 groups shown in panel C ($F_{2,23}$ = 4.90, p<0.05, one-way ANOVA). *p<0.05 (Bonferroni post hoc test). None of the treatments in this figure affected the overall activity level during the pretest (*Figure 7—figure supplement 2*).

The following figure supplements are available for figure 7:

**Figure supplement 1.** Unhandled control rats developed robust CPP after 3-day conditioning with cocaine (5 mg/kg).

**Figure supplement 2.** Summary graphs depicting the overall activity level (i.e., total number of beam breaks in the CPP box compartments) during the pretest for the experiments shown in *Figure 7*.

**Figure supplement 3.** Photograph of CPP box compartment with color-contrasting ceramic weight.

## Repeated social stress promotes learning of cocaine-associated cues in a GR-dependent manner

Next, the effect of social defeat stress was tested on acquisition of cocaine CPP, in which animals learn to associate a particular context with drug reward. Acquisition of psychostimulant CPP is

inhibited by mGluR1 or NMDAR antagonist in the VTA, while CPP expression is attenuated by NMDAR antagonist, but not by mGluR1 antagonist, in the VTA (*Whitaker et al., 2013*), supporting the potential role of NMDAR LTP in driving CPP. Rats underwent stress or control procedures for 5 days, then underwent 1-day CPP conditioning with cocaine (5 mg/kg, i.p.). It should be noted that a single psychostimulant treatment does not cause $IP_3R$ sensitization (*Ahn et al., 2010*; *Whitaker et al., 2013*). Stressed rats displayed robust preference for the cocaine-paired side after 1-day conditioning, while unhandled and handled controls showed small preference (*Figure 7A*). The 1-day CPP score was significantly larger in stressed rats compared to unhandled and handled controls (*Figure 7B*). Control rats developed significant cocaine CPP comparable to that observed in stressed rats after 3-day conditioning with the same dose of cocaine (*Figure 7—figure supplement 1*). These data suggest that repeated defeat experience promotes the rate of learning of cocaine-associated cues.

Finally, we asked whether GR signaling, which is necessary for $IP_3R$ sensitization, also plays a role in promoting cocaine CPP. As in the electrophysiology experiments, rats were treated with mifepristone (40 mg/kg, i.p.) or vehicle 30 min before each social defeat session. An additional group received mifepristone followed by control handling procedure. We found that mifepristone suppressed cocaine CPP in stressed rats to a level comparable to that observed in mifepristone-treated controls (*Figure 7C and D*). Therefore, GR activation during stress is required for CPP enhancement.

## Discussion

Repeated stressful experience leads to metaplasticity, i.e., experience-dependent changes in the capacity of synapses to undergo activity-dependent plasticity (*Abraham, 2008*), in different brain areas (*Kim and Diamond, 2002*; *Joels et al., 2006*; *Schwabe et al., 2012*; *Chattarji et al., 2015*). The present study demonstrates that repeated social defeat facilitates the induction of LTP of NMDAR-mediated transmission in VTA dopamine neurons while causing no alterations in global NMDAR-mediated excitation or intrinsic firing activity. Importantly, socially defeated animals display enhanced acquisition of cocaine CPP, a form of Pavlovian conditioning that requires NMDAR-dependent bursting in the VTA (*Zweifel et al., 2009*; *Wang et al., 2011*; *Whitaker et al., 2013*).

Repeated social defeat results in increased sensitivity of $IP_3Rs$, which serve as a coincidence detector of presynaptic activity (causing mGluR-dependent $IP_3$ generation) and postsynaptic bursting (driving $Ca^{2+}$ influx) during NMDAR LTP induction (*Harnett et al., 2009*). Inhibition of PKA completely reversed the increase in the potency of $IP_3$, indicating the role of PKA-dependent phosphorylation in stress-induced $IP_3R$ sensitization, as has been suggested in previous studies demonstrating similar changes following repeated drug exposure (*Ahn et al., 2010*; *Bernier et al., 2011*). It is of note that dopamine neurons in the substantia nigra pars compacta, in contrast to VTA neurons recorded in the present study, display significant PKA-dependent regulation of $IP_3$-induced $Ca^{2+}$ signaling and NMDAR LTP induction in control rats, which cannot be further enhanced by repeated drug exposure (*Harnett et al., 2009*; *Ahn et al., 2010*).

It has been reported that repeated social defeat (10 days) in mice leads to long-lasting (>4 weeks) alterations in gene expression and behavior (e.g., reduced social contact), with little recovery unless treated with antidepressants (*Berton et al., 2006*; *Tsankova et al., 2006*). In the present study, mGluR/$IP_3$ action on burst-evoked $Ca^{2+}$ signals remained elevated for 10–30 days following the 5-day defeat paradigm in rats, although displaying gradual decline during the 30-day stress-free period. It remains to be determined how this recovery is affected by different stress paradigms (e.g., duration or type/severity) and treatments following stress experience.

Mouse studies have also shown that individual animals display different susceptibility to repeated social defeat when assessed by the degree of social avoidance, which correlates with biochemical and physiological changes in the mesolimbic system (*Krishnan et al., 2007*; *Cao et al., 2010*). In particular, these studies observed hyperactivity of VTA dopamine neurons, assessed in vivo or ex vivo, associated with an increase in $I_h$, after 10-day defeat in susceptible mice. However, these parameters were not affected by 5-day social defeat in the current study conducted in rats.

How could repeated stress lead to increased PKA activity regulating $IP_3Rs$ in the VTA? Stress, including social defeat, promotes bursting in a subpopulation of VTA dopamine neurons, causing increased dopamine transients in the nucleus accumbens (*Anstrom et al., 2009*; *Brischoux et al., 2009*). Bursting also releases dopamine locally from the soma and dendrites, activating

somatodendritic D2 autoreceptors (*Beckstead et al., 2004*). Chronic stimulation of $G_i$-coupled receptors, such as D2 receptors, is known to upregulate the cAMP-PKA pathway (*Hyman et al., 2006*), which may contribute to stress-induced $IP_3R$ sensitization. Interestingly, intra-VTA blockade of NMDARs during each social defeat episode, which would suppress dopamine neuron bursting, has been shown to prevent repeated stress-induced increases in cocaine self-administration (*Covington et al., 2008*).

GR signaling during defeat sessions is necessary for the enhancement of $IP_3R$ sensitivity. Stress-induced activation of the mesolimbic dopamine system is regulated by glucocorticoids (*Marinelli and Piazza, 2002*). Recent evidence implicates GRs expressed in projection areas, not in the VTA, in long-term glucocorticoid regulation of dopamine neuron activity (*Ambroggi et al., 2009*; *Butts et al., 2011*; *Barik et al., 2013*). Furthermore, glucocorticoids can also enhance synthesis of corticotropin-releasing factor (CRF), a major stress-related neuropeptide, and activation of CRF neurons in brain areas providing major CRF inputs to the VTA (*Makino et al., 1994*; *Rodaros et al., 2007*; *Kolber et al., 2008*). GR blockade may therefore attenuate CRF-induced excitation of dopamine neurons during stress (*Kalivas et al., 1987*; *Ungless et al., 2003*; *Wanat et al., 2008*; *Holly et al., 2015*). We found that GR activation alone is not sufficient for $IP_3R$ sensitization. Thus, the potential GR mechanisms described above may act to amplify glutamatergic input-driven bursting activity during stress episodes, likely further enhanced by stress-induced activation of noradrenergic inputs stimulating dopamine neurons via $\alpha_1$ adrenergic receptors (*Grenhoff et al., 1995*; *Paladini et al., 2001*; *Morilak et al., 2005*), thereby enabling large local dopamine release in the VTA. In this regard, it is interesting that repeated cocaine treatment was capable of causing similar enhancement of $mGluR/IP_3$ action in a GR-independent manner. Dopamine levels in the VTA caused by cocaine alone are likely sufficient to induce D2-mediated upregulation of the cAMP-PKA pathway.

Increased $IP_3R$ sensitivity drives the enhancement of NMDAR LTP induction in socially defeated animals. Our recent study demonstrated the involvement of L-type $Ca^{2+}$ channels (LTCCs) in NMDAR LTP (*Degoulet et al., 2015*). Although glucocorticoid-induced upregulation of LTCCs has been reported in the hippocampus and amygdala (*Karst et al., 2002*; *Chameau et al., 2007*), pharmacological activation of these channels does not enhance NMDAR LTP in dopamine neurons (*Degoulet et al., 2015*); thus changes in LTCCs are unlikely to play a role in LTP enhancement.

CPP experiments showed that repeated social defeat promoted acquisition of the preference for contextual cues paired with cocaine experience, in accordance with previous studies demonstrating enhanced drug CPP following a period of repeated stress (*Kreibich et al., 2009*; *Burke et al., 2011*). Blockade of the critical components regulating NMDAR LTP induction (i.e., NMDARs, group I mGluRs, PKA, or LTCCs) in the VTA during conditioning has been shown to suppress CPP acquisition (*Harnett et al., 2009*; *Ahn et al., 2010*; *Whitaker et al., 2013*; *Degoulet et al., 2015*). In the present study, systemic GR blockade during defeat episodes prevented both the enhancement of the LTP induction mechanism and that of cocaine CPP acquisition, consistent with the potential role of NMDAR plasticity in this form of Pavlovian learning. However, enhanced CPP acquisition observed in defeated rats may well be caused by an increase in the primary rewarding action of cocaine itself. The relative contribution of these two possibilities, i.e., enhanced learning mechanism vs. enhanced cocaine reward, remains to be determined.

It is well known that repeated stress impairs LTP of AMPAR-mediated transmission in the hippocampus (*Foy et al., 1987*; *Shors et al., 1989*), an effect that requires GR activation during stress (*Xu et al., 1998*). LTP is similarly impaired in the prefrontal cortex (*Goldwater et al., 2009*). By contrast, repeated stress leads to enhancement of AMPAR LTP in the lateral amygdala, which underlies Pavlovian fear conditioning driven by stressful/aversive stimuli (*Rodriguez Manzanares et al., 2005*; *Suvrathan et al., 2014*). Alterations in the function/expression of NMDARs are implicated in these forms of metaplasticity, as NMDARs play a key role in AMPAR LTP induction (*Kim and Diamond, 2002*; *Chattarji et al., 2015*). Here, we described a distinct form of stress-induced metaplasticity in the VTA, i.e., enhancement of $mGluR/IP_3$-dependent NMDAR LTP, which may, at least in part, contribute to the enhanced drug reward-based Pavlovian learning. This may illuminate a key mechanism by which stressful experience increases vulnerability to addiction, a chronic relapsing disorder perpetuated by memories of drug-associated stimuli.

# Materials and methods

### Animals

Sprague-Dawley rats (Harlan Laboratories, Houston, Texas) were housed in groups of 2–3 on a 12 hr light/dark cycle with food and water available ad libitum. All procedures were approved by the University of Texas Institutional Animal Care and Use Committee.

### Resident-intruder social defeat paradigm

Twelve week-old male resident rats were vasectomized and pair-housed with 6 week-old females. Residents (used for ~8–10 months) were screened for aggression (biting or pinning within 1 min) by introducing a male intruder to the home cage. Intruders and controls were young males (4–5 weeks old at the beginning) housed in groups of 2–3. For defeat sessions, residents and intruders were taken to a darkened procedure room at the end of the dark cycle. Intruders were introduced to residents' home cages after removing females. Following 5 min of direct contact, a perforated Plexiglass barrier was inserted for 25 min to allow sensory contact. For repeated defeat, intruders underwent one session daily with a novel resident. Handled controls were taken to a darkened procedure room and placed in novel cages for 30 min. Unhandled controls remained undisturbed in the colony. Intruders and controls were housed separately.

### In vivo drug treatments

All drug and vehicle solutions were administered via i.p. injections (1 ml/kg). Mifepristone and corticosterone (both from Tocris Bioscience, Ellisville, Missouri) were dissolved in 30% propylene glycol plus 1% Tween-20 in 0.9% saline. Cocaine-HCl (Sigma-Aldrich, St. Louis, Missouri) was dissolved in 0.9% saline.

### Electrophysiology

Midbrain slices were prepared and recordings were made in the lateral VTA located 50–150 µm from the medial border of the medial terminal nucleus of the accessory optic tract, as in our previous studies (*Ahn et al., 2010*; *Whitaker et al., 2013*; *Degoulet et al., 2015*). Tyrosine hydroxylase-positive neurons in this area (i.e., lateral part of the parabrachial pigmented nucleus) largely project to the ventrolateral striatum (*Ikemoto, 2007*) and show little VGluT2 coexpression (*Trudeau et al., 2014*). Putative dopamine neurons in the lateral VTA were identified by spontaneous firing of broad APs (>1.2 ms) at 1–5 Hz in cell-attached configuration and large $I_h$ currents (>200 pA; evoked by a 1.5 s hyperpolarizing step of 50 mV) in whole-cell configuration (*Ford et al., 2006*; *Lammel et al., 2008*; *Margolis et al., 2008*). Cells were voltage-clamped at –62 mV (corrected for –7 mV liquid junction potential).

A 2 ms depolarizing pulse of 55 mV was used to elicit an unclamped AP. For bursts, 5 APs were evoked at 20 Hz. The time integral of the outward tail current, termed $I_{K(Ca)}$ (calculated after removing the 20 ms window following each depolarizing pulse; expressed in pC), was used as a readout of AP-evoked $Ca^{2+}$ transients, as it is eliminated by TTX and also by apamin, a blocker of $Ca^{2+}$-activated SK channels (*Cui et al., 2007*).

### Flash photolysis

Cells were loaded with caged $IP_3$ (50–400 µM; generous gift from Dr. Kamran Khodakhah) through the recording pipette. A UV flash (~1 ms) was applied with a xenon arc lamp driven by a photolysis system (Cairn Research, Faversham, UK). The UV flash was focused through a 60× objective onto a ~350 µm area surrounding the recorded neuron. Photolysis of caged compounds is proportional to the UV flash intensity; therefore, the concentration of $IP_3$ was defined as the product of caged $IP_3$ concentration in the pipette (µM) and flash intensity (µJ) measured at the focal plane of the objective (expressed in µM·µJ).

### NMDAR LTP experiments

Synaptic stimuli were delivered with a bipolar tungsten electrode placed ~50–100 µm rostral to the recorded neuron. To isolate NMDAR EPSCs, recordings were performed in DNQX (10 µM), picrotoxin (100 µM), CGP54626 (50 nM), and sulpiride (100 nM) to block AMPA/kainate, $GABA_A$, $GABA_B$,

and $D_2$ dopamine receptors, and in glycine (20 μM) and low $Mg^{2+}$ (0.1 mM) to enhance NMDAR activation. NMDAR EPSCs were monitored every 20 s. The LTP induction protocol consisted of photolytic application of $IP_3$ (250 μM·μJ) 50 ms prior to the simultaneous delivery of synaptic stimulation (20 stimuli at 50 Hz) and a burst (5 APs at 20 Hz), repeated 10 times every 20 s. LTP magnitude was determined by comparing the average EPSC amplitude 30 min post-induction with the average EPSC amplitude pre-induction (each from a 5 min window).

### Place conditioning

A CPP box (Med Associates, St. Albans, Vermont) consisting of two distinct compartments separated by a small middle chamber was used for conditioning. One compartment had a mesh floor with white walls, while the other had a grid floor with black walls. A discrete cue (painted ceramic weight) was placed in the rear corner of each compartment (black one in the white wall side, white one in the black wall side; *Figure 7—figure supplement 3*) for further differentiation. One day after undergoing repeated stress or control procedures, rats were pretested for initial side preference by exploring the entire CPP box for 15 min. The percentage of time spent in each compartment was determined after excluding the time spent in the middle chamber. Rats with initial side preference >60% were excluded. Starting the next day, rats were subjected to 1-day or 3-day conditioning, in which they were given a saline injection in the morning and confined to one compartment, then in the afternoon given cocaine (5 mg/kg) and confined to the other compartment (10 min each). Compartment assignment was counterbalanced such that animals had, on average, ~50% initial preference for the cocaine-paired side. A 15 min posttest was performed 1 day after the last conditioning session. The CPP score was determined by subtracting the preference for the cocaine-paired side during pretest from that during posttest. The experimenter performing CPP experiments was blind to animal treatments.

### Data analysis

Data are expressed as mean ± SEM. Statistical significance was determined by Student's *t*-test or ANOVA followed by Bonferroni *post hoc* test. Normality of data distribution was confirmed by Kolmogorov-Smirnov test. The difference was considered significant at p<0.05.

## Acknowledgements

This work was supported by NIH grants DA015687 and AA015521. JBC was supported by NIH training grant AA007471. We thank Dr. Kamran Khodakhah for the generous gift of caged $IP_3$ made in his lab at Albert Einstein College of Medicine. We also thank Dr. Michela Marinelli for comments on this manuscript and Dr. Kevin Lominac and Nhi Le for assistance with CPP experiment.

## Additional information

### Funding

| Funder | Grant reference number | Author |
|---|---|---|
| National Institutes of Health | AA007471 | Jason B Cook |
| National Institutes of Health | DA015687 | Hitoshi Morikawa |
| National Institutes of Health | AA015521 | Hitoshi Morikawa |

The funders had no role in study design, data collection and interpretation, or the decision to submit the work for publication.

### Author contributions

CES, Conception and design, Acquisition of data, Analysis and interpretation of data, Drafting or revising the article; MBP, JBC, Acquisition of data, Analysis and interpretation of data; HM, Conception and design, Analysis and interpretation of data, Drafting or revising the article

### Author ORCIDs

Hitoshi Morikawa, http://orcid.org/0000-0002-2948-493X

## Ethics

Animal experimentation: All animal procedures were approved by the University of Texas Institutional Animal Care and Use Committee (Protocol ID: AUP-2013-00177).

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
