## [Decision Letter]

Thank you for submitting your article "Repeated social defeat stress enhances synaptic plasticity that drives learning of drug-associated cue valence" for consideration by *eLife*. Your article has been reviewed by three peer reviewers, and the evaluation has been overseen by a Reviewing Editor, Naoshige Uchida, and a Senior Editor. One of the three reviewers has agreed to share his identity: Jochen Roeper.

The reviewers have discussed the reviews with one another and the Reviewing Editor has drafted this decision to help you prepare a revised submission.

Summary:

This study found that repeated social defeat stress enhances long-term potentiation of NMDA receptor-mediated glutamate transmission in the ventral tegmental area (VTA). The authors also found that protein kinase A-dependent sensitization of IP3-mediated calcium release plays a key role in this process, and that this plasticity is triggered by stress-related glucocortcoids (GR). The authors also present evidence indicating that cocaine-induced conditioned place preference (CPP) conditioning is enhanced after repeated social defeat stress. The referees found this study of significance and timely, addressing important questions relevant to the mechanism underlying drug addiction. The manuscript is well written and the data are generally sound. However, there are two major issues upon which we would like to see your response.

Essential revisions:

1) It was pointed out that the CPP experiment is difficult to interpret without further control data. Specifically, the current protocol (1 day conditioning with 5mg/kg cocaine) does not induce CPP robustly in control animals (Figure 7). This may be because the dose of cocaine used, together with the other experimental conditions, is not rewarding at all, or because conditioning was not enough. Depending on these possibilities, the interpretation of the result can be very different. If the authors claim that social defeat stress enhanced learning, the authors should show that this dose of cocaine is rewarding, and extended conditioning actually causes CPP in control mice. Thus, we would like to see the data indicating that positive CPP can be observed in control animals with the same dose of cocaine.

2) The authors interpret the result as "enhanced learning of associated cue valence", however, it is possible that the increased CPP is due to an enhanced rewarding effect of cocaine. Please interpret the results more carefully so that it becomes clearer that the latter interpretation is plausible.

Reviewer #1:

This paper is well written and effectively discusses a series a thoughtful experiments surrounding the authors' central question. Namely, that GR receptors inDA neurons of the VTA shape changes in synaptic strength following an ethologically relevant model of stress and this is accompanied by enhanced place conditioning for cocaine. However, there are major issues that should be addressed prior to publication.

Perhaps the most serious issue, in my opinion, is the interpretation of the CPP data. Here the authors observe a CPP in stressed rats after a single conditioning trial, however, a similar CPP is not developed in control animals.

1) Without a positive control (i.e. an observed CPP in controls), these data are not interpretable. I understand the journal's mandate to not require additional experiments, but this is one of those cases where this is critical. Are CPPs generally observed after a single conditioning session with a 5mg/kg IP dose? If not, the authors should extend the conditioning protocol until a CPP can be observed in all animals.

2) The authors interpret CPP as "enhanced learning of associated cue valence", however, it could very well also be just enhanced reward of cocaine. CPP technically measures reward, so it is unclear why the authors did not acknowledge the possibility (in my opinion, strong likelihood) that social defeat stress actually enhances cocaine reward rather than Pavlovian associations. Please revise thoroughly to account for this alternative possibility.

3) The authors overstate the involvement of dopamine in learning – this is a well-supported theory, but certainly only one of several. In the Introduction the authors state, "…dopamine neuron responses drive the learning of Pavlovian cue-reward associations", this should be adjusted to reflect that this is a theory, perhaps, " dopamine neuron responses are hypothesized to drive the learning…".

4) The involvement of LTP in the CPP behavioral data is too speculative, for example in the first paragraph of the subsection “Repeated Social Stress Promotes Learning of Cocaine-Associated Cues in a GR-Dependent Manner”; please revise. Specifically, there is not a single piece of empirical evidence that the observed LTP "promotes" the Pavlovian associations that give to rise to CPP. This should be thoroughly re-written, to reflect the findings as they actually are. Namely, that LTP enhancement accompanies social defeat stress (or it could in fact be an epiphenomenon of the defeat). To show unambiguously that it "promotes" the behavioral effects observed, would require a whole new set of experiments, which fall outside of the scope of this study.

5) Animal/group numbers were poorly described. Please clarify these as well as the degrees of freedom for all experiments.

Reviewer #2:

This is an excellent study which has discovered that chronic social defeat stress in rats leads to a persistent enhancement of in vitro NMDA-LTP in DA VTA neurons, without changing intrinsic firing properties. The Morikawa lab has previously discovered and carefully characterized this type of plasticity. Here, they also show that PKA-dependent sensitization of IP3-mediated calcium release is a key component of the underlying signalling process and that this plasticity is triggered by stress-related glucocortcoids (GR). Importantly, they give also some evidence that the GR-dependent enhanced NMDA-LTP might be relevant for potentiating cocaine-induced condition place-preference learning. The behavioral data do however not prove that NMDA-LTP itself in causal for the learning improvement, but it a likely candidate.

While the Discussion covers well the changes in signaling processes induced by stress that might converge on enhanced NMDA-LTP, it avoids a very important issue regarding the different outcome of chronic social defeat stress (resilient and non-resilient) and the correlative and even causal link to changes in DA VTA neuronal firing. In contrast, this study sees no changes in DA VTA firing, which might imply that these were highly resilient rats. It would be important to provide additional data on the post-stress behaviors of the rats used, e.g. reduced social approach and/or anhedonia that have been associated with altered in vivo and vitro firing pattern of DA VTA – enhanced mean frequencies and increased bursting. Interestingly, the molecular mechanism identified here would be a very good candidate for inducing these activity pattern in DA VTA neurons of non-resilient post-defeat animals. In addition to providing this information, the authors should more actively engage with this prominent literature.

Reviewer #3:

In general, this is an interesting manuscript. The data is sound, and the central message is timely and of significance.

1) If recruitment of Ca^2+^-activated K^+^ currents is enhanced after stress, then there may be a change in firing pattern and waveform (i.e. AHP) of glutamate-evoked spiking of DA neurons, but not tonic as observed. Did the authors consider examining this measure?

2) Is the enhancement of IP3 sensitivity after stress similar for all Gq-coupled GPCRs? Or is only mGluR1 function increased due to the IP3 enhancement? This may have important implications for understanding functional consequences.

3) An additional experiment with a more direct link between the cocaine CPP and changes in IP3 signaling in VTA DA neurons after stress would help strengthen the study.

[Editors' note: further revisions were requested prior to acceptance, as described below.]

Thank you for submitting your article "Repeated social defeat stress enhances synaptic plasticity that drives learning of drug-associated cue valence" for consideration by *eLife*. Your article has been reviewed by three peer reviewers, and the evaluation has been overseen by a Reviewing Editor, Naoshige Uchida, and a Senior Editor. One of the three reviewers has agreed to share his identity: Jochen Roeper.(Reviewer #2).

The reviewers have discussed the reviews with one another and the Reviewing Editor has drafted this decision to help you prepare a revised submission.

Summary:

While the reviewers appreciated that the addition of the new conditioned place preference (CPP) experiment has greatly improved the manuscript, reviewer #1 remained unsatisfied by the authors' interpretations.

Essential revisions:

Specifically, the authors should discuss, in a more balanced way, the possibility that cocaine exposures enhanced rewarding effects of cocaine not merely the learning of it. Additionally, the referee remained unsatisfactory regarding the discussion on the link between the LTP and CPP behavioral data. The discussion must be more explicit that a causal demonstration will be required to establish the link.

The reviewers agreed that these are interpretational issues and do not require additional experiments. Nonetheless, the manuscript will greatly benefit from discussing these in a more careful way. For more details, please refer to the referee's comments that are copied below:

Reviewer #1:

The current version of the manuscript is improved by the addition of the new CPP results. However, issues still remain with the authors' interpretation of their data that make this manuscript incomplete.

1) I remain unsatisfied that the CPP protocol utilized here measured learning of cocaine-associated cues and not cocaine reward.

The authors now acknowledge that CPP measures reward in the eighth paragraph of the Discussion, however, this is quickly overshadowed by a qualifying statement which is contradictory:

"Enhanced CPP acquisition observed in defeated rats might be due to an increase in the "primary rewarding" action of cocaine. Although this possibility cannot be ruled out, it should be noted that socially defeated rats displayed no significant increases in the overall activity level during the cocaine conditioning sessions (Figure 7—figure supplement 1), suggesting that the "locomotor stimulating" action of cocaine was not altered by repeated defeat experience."

This statement is very misleading. Are the authors suggesting that because stressed mice did not show enhanced cocaine locomotor activity compared to control mice, that they likely did not have enhanced cocaine reward? If in fact this is what the authors are trying to convey, then I would surmise that they are equating locomotor activation with reward, which has repeatedly been shown not to be the case (for example, see PMID: 22197517).

As such, the interpretation of the data continues to the main problem of the paper. Place conditioning is a test of reward and/or aversion and not a test that provides a clear examination of drug-associated cues without more specific controls in place. Perhaps if the authors had administered drug in the CPP chamber or, better yet, had the animals self-administer the drug in the CPP chamber then this interpretation would be more acceptable. As is, the question they are answering is one of subjective cocaine experience and not of learning cues associated with drug taking.

In fact, the citations the authors employ to justify their use of CPP clearly state that CPP is a measure of reward: "…Pavlovian cue-outcome associations driven by rewarding stimuli, assessed with conditioned place preference (CPP) (Kreibich et al., 2009; Burke et al., 2011; Chuang et al., 2011)…".

Kreibich AS, Briand L, Cleck JN, Ecke L, Rice KC, Blendy JA (2009) Stress-induced potentiation of cocaine reward: a role for CRF R1 and CREB. Neuropsychopharmacology 34:2609-2617.

Burke AR, Watt MJ, Forster GL (2011) Adolescent social defeat increases adult amphetamine conditioned place preference and alters D2 dopamine receptor expression. Neuroscience 197:269-279.

"The effects of adolescent stress on drug preferences in adulthood were studied."

Chuang JC, Perello M, Sakata I, Osborne-Lawrence S, Savitt JM, Lutter M, Zigman JM (2011) Ghrelin mediates stress-induced food-reward behavior in mice. The Journal of clinical investigation 121:2684-2692.

Moreover, the authors refer to the cocaine chamber as cocaine-associated cues, however, these contextual cues are in fact paired with the experience of cocaine and not cocaine administration itself, which, presumably, takes place elsewhere. This should be made clear. It is as if the authors lack some of the fundamental knowledge of Pavlovian conditioning or choose to ignore it to fit the framework of the paper. For example:

"Repeated social defeat enhanced learning of cocaine-associated cues assessed with a CPP paradigm."

2) The authors have done little to assuage initial concerns that the involvement of LTP in the CPP behavioral data is greatly overstated.

The data clearly show that social stress enhances LTP and this is glucocorticoid-dependent. Enhancement of CPP seen in stressed rats is also glucocorticoid-dependent. These are important findings and the experiments are performed elegantly. However, there is no assessment that unambiguously shows that enhanced LTP is required for the observed facilitation of cocaine CPP. As such, there are a large number of instances throughout the manuscript wherein the authors overstate their findings (as was mentioned in the original review).

For example, the title of this manuscript is unsupported by its contents:

"Repeated social defeat stress enhances synaptic plasticity that drives learning of drug- associated cue valence"

The authors do not provide any empirical data to support that LTP drives CPP.

In the Abstract, this trend continues when the authors state:

"Here, we show that repeated social defeat stress in rats results in persistent enhancement of long-term potentiation (LTP) of NMDA receptor-mediated glutamatergic transmission in the ventral tegmental area (VTA), thereby promoting the learning of cocaine-associated contextual cues assessed with a conditioned place preference (CPP) paradigm."

and…

"…plasticity in the VTA as the critical mechanism by which repeated stress increases addiction vulnerability."

Likewise, statements such as the following are far too speculative (i.e., not a single piece of evidence is provided to support this claim):

"Here, mGluR1/NMDAR blockade would suppress CPP acquisition via inhibiting LTP induction at glutamatergic inputs activated by contextual cues of the CPP box, while blocking potentiated NMDAR-mediated excitation at those inputs would interfere with CPP expression".

Reviewer #2:

I am happy with the revised version and believe that all essential points have been addressed adequately.

Reviewer #3:

I understand the reluctance of the authors to add many additional experiments.

With that in mind, I am OK for this article to be accepted for publication, should the other reviewers feel the same.

---

## [Author Response]

*Essential revisions:*

1) It was pointed out that the CPP experiment is difficult to interpret without further control data. Specifically, the current protocol (1 day conditioning with 5mg/kg cocaine) does not induce CPP robustly in control animals (Figure 7). This may be because the dose of cocaine used, together with the other experimental conditions, is not rewarding at all, or because conditioning was not enough. Depending on these possibilities, the interpretation of the result can be very different. If the authors claim that social defeat stress enhanced learning, the authors should show that this dose of cocaine is rewarding, and extended conditioning actually causes CPP in control mice. Thus, we would like to see the data indicating that positive CPP can be observed in control animals with the same dose of cocaine.

We have performed 3-day CPP conditioning with 5 mg/kg cocaine in both control and socially defeated rats (Figure 7—figure supplement 1). We found that control rats displayed robust CPP comparable to defeated rats when the number of conditioning sessions was increased from one to three.

2) The authors interpret the result as "enhanced learning of associated cue valence", however, it is possible that the increased CPP is due to an enhanced rewarding effect of cocaine. Please interpret the results more carefully so that it becomes clearer that the latter interpretation is plausible.

This possibility is explicitly discussed in the eighth paragraph of the Discussion.

*Reviewer #1:*

*This paper is well written and effectively discusses a series a thoughtful experiments surrounding the authors' central question. Namely, that GR receptors inDA neurons of the VTA shape changes in synaptic strength following an ethologically relevant model of stress and this is accompanied by enhanced place conditioning for cocaine. However, there are major issues that should be addressed prior to publication.*

*Perhaps the most serious issue, in my opinion, is the interpretation of the CPP data. Here the authors observe a CPP in stressed rats after a single conditioning trial, however, a similar CPP is not developed in control animals.*

1) Without a positive control (i.e. an observed CPP in controls), these data are not interpretable. I understand the journal's mandate to not require additional experiments, but this is one of those cases where this is critical. Are CPPs generally observed after a single conditioning session with a 5mg/kg IP dose? If not, the authors should extend the conditioning protocol until a CPP can be observed in all animals.

This issue is discussed above in Essential revisions: item 1.

2) The authors interpret CPP as "enhanced learning of associated cue valence", however, it could very well also be just enhanced reward of cocaine. CPP technically measures reward, so it is unclear why the authors did not acknowledge the possibility (in my opinion, strong likelihood) that social defeat stress actually enhances cocaine reward rather than Pavlovian associations. Please revise thoroughly to account for this alternative possibility.

This issue is addressed in Essential revisions: item 2.

3) The authors overstate the involvement of dopamine in learning – this is a well-supported theory, but certainly only one of several. In the Introduction the authors state, "…dopamine neuron responses drive the learning of Pavlovian cue-reward associations", this should be adjusted to reflect that this is a theory, perhaps, " dopamine neuron responses are hypothesized to drive the learning…".

This sentence was revised as suggested.

4) The involvement of LTP in the CPP behavioral data is too speculative, for example in the first paragraph of the subsection “Repeated Social Stress Promotes Learning of Cocaine-Associated Cues in a GR-Dependent Manner”; please revise. Specifically, there is not a single piece of empirical evidence that the observed LTP "promotes" the Pavlovian associations that give to rise to CPP. This should be thoroughly re-written, to reflect the findings as they actually are. Namely, that LTP enhancement accompanies social defeat stress (or it could in fact be an epiphenomenon of the defeat). To show unambiguously that it "promotes" the behavioral effects observed, would require a whole new set of experiments, which fall outside of the scope of this study.

We have revised the parts describing the potential role of NMDAR LTP in the VTA in CPP (subsection “Repeated Social Stress Promotes Learning of Cocaine-Associated Cues in a GR-Dependent 178 Manner”, first paragraph; Discussion, last paragraph).

5) Animal/group numbers were poorly described. Please clarify these as well as the degrees of freedom for all experiments.

These types of information are all provided in the figure legends.

Reviewer #2:

*While the Discussion covers well the changes in signaling processes induced by stress that might converge on enhanced NMDA-LTP, it avoids a very important issue regarding the different outcome of chronic social defeat stress (resilient and non-resilient) and the correlative and even causal link to changes in DA VTA neuronal firing. In contrast, this study sees no changes in DA VTA firing, which might imply that these were highly resilient rats. It would be important to provide additional data on the post-stress behaviors of the rats used, e.g. reduced social approach and/or anhedonia that have been associated with altered* in vivo

We have added a paragraph addressing this issue (Discussion, fourth paragraph).

*Reviewer #3:*

*In general, this is an interesting manuscript. The data is sound, and the central message is timely and of significance.*

1) If recruitment of Ca^2+^-activated K^+^ currents is enhanced after stress, then there may be a change in firing pattern and waveform (i.e. AHP) of glutamate-evoked spiking of DA neurons, but not tonic as observed. Did the authors consider examining this measure?

The size of the SK current evoked by bursts, termed basal burst-evoked I_K(Ca)_, was not altered by repeated social defeat stress (Figure 1).

2) Is the enhancement of IP3 sensitivity after stress similar for all Gq-coupled GPCRs? Or is only mGluR1 function increased due to the IP3 enhancement? This may have important implications for understanding functional consequences.

We do think that IP_3_ signaling evoked by all Gq-coupled receptors would be affected and we agree that this would have important implications. We appreciate this insightful comment.

*3) An additional experiment with a more direct link between the cocaine CPP and changes in IP3 signaling in VTA DA neurons after stress would help strengthen the study.*

Although directly manipulating IP_3_ signaling in the VTA in vivo would not be a simple experiment, again we appreciate this insightful comment for our future study.

[Editors' note: further revisions were requested prior to acceptance, as described below.]

*Essential revisions:*

*Specifically, the authors should discuss, in a more balanced way, the possibility that cocaine exposures enhanced rewarding effects of cocaine not merely the learning of it. Additionally, the referee remained unsatisfactory regarding the discussion on the link between the LTP and CPP behavioral data. The discussion must be more explicit that a causal demonstration will be required to establish the link.*

*The reviewers agreed that these are interpretational issues and do not require additional experiments. Nonetheless, the manuscript will greatly benefit from discussing these in a more careful way. For more details, please refer to the referee's comments that are copied below:*

*Reviewer #1:*

*The current version of the manuscript is improved by the addition of the new CPP results. However, issues still remain with the authors' interpretation of their data that make this manuscript incomplete.*

*1) I remain unsatisfied that the CPP protocol utilized here measured learning of cocaine-associated cues and not cocaine reward.*

*The authors now acknowledge that CPP measures reward in the eighth paragraph of the Discussion, however, this is quickly overshadowed by a qualifying statement which is contradictory:*

*"Enhanced CPP acquisition observed in defeated rats might be due to an increase in the "primary rewarding" action of cocaine. Although this possibility cannot be ruled out, it should be noted that socially defeated rats displayed no significant increases in the overall activity level during the cocaine conditioning sessions (Figure 7—figure supplement 1), suggesting that the "locomotor stimulating" action of cocaine was not altered by repeated defeat experience."*

This statement is very misleading. Are the authors suggesting that because stressed mice did not show enhanced cocaine locomotor activity compared to control mice, that they likely did not have enhanced cocaine reward? If in fact this is what the authors are trying to convey, then I would surmise that they are equating locomotor activation with reward, which has repeatedly been shown not to be the case (for example, see PMID: 22197517).

We have removed this statement and replaced it with the following statement:

“However, enhanced CPP acquisition observed in defeated rats may well be caused by an increase in the primary rewarding action of cocaine itself. The relative contribution of these two possibilities, i.e., enhanced learning mechanism vs. enhanced cocaine reward, remains to be determined.”

*Moreover, the authors refer to the cocaine chamber as cocaine-associated cues, however, these contextual cues are in fact paired with the experience of cocaine and not cocaine administration itself, which, presumably, takes place elsewhere. This should be made clear. It is as if the authors lack some of the fundamental knowledge of Pavlovian conditioning or choose to ignore it to fit the framework of the paper. For example:*

"Repeated social defeat enhanced learning of cocaine-associated cues assessed with a CPP paradigm."

This sentence has been rewritten as follows:

“CPP experiments showed that repeated social defeat promoted acquisition of the preference for contextual cues paired with cocaine experience.”

*2) The authors have done little to assuage initial concerns that the involvement of LTP in the CPP behavioral data is greatly overstated.*

*The data clearly show that social stress enhances LTP and this is glucocorticoid-dependent. Enhancement of CPP seen in stressed rats is also glucocorticoid-dependent. These are important findings and the experiments are performed elegantly. However, there is no assessment that unambiguously shows that enhanced LTP is required for the observed facilitation of cocaine CPP. As such, there are a large number of instances throughout the manuscript wherein the authors overstate their findings (as was mentioned in the original review).*

*For example, the title of this manuscript is unsupported by its contents:*

"Repeated social defeat stress enhances synaptic plasticity that drives learning of drug- associated cue valence"

The title has been changed as follows:

“Repeated social defeat stress enhances glutamatergic synaptic plasticity in the VTA and cocaine place conditioning”.

*In the Abstract, this trend continues when the authors state:*

*"Here, we show that repeated social defeat stress in rats results in persistent enhancement of long-term potentiation (LTP) of NMDA receptor-mediated glutamatergic transmission in the ventral tegmental area (VTA), thereby promoting the learning of cocaine-associated contextual cues assessed with a conditioned place preference (CPP) paradigm."*

*and…*

"…plasticity in the VTA as the critical mechanism by which repeated stress increases addiction vulnerability."

The Abstract has been rewritten in two places as follows:

“Notably, defeated rats display enhanced learning of contextual cues paired with cocaine experience assessed using a conditioned place preference (CPP) paradigm.”

“These findings suggest that enhanced glutamatergic plasticity in the VTA may contribute, at least partially, to increased addiction vulnerability following repeated stressful experiences.”

*Likewise, statements such as the following are far too speculative (i.e., not a single piece of evidence is provided to support this claim):*

*"Here, mGluR1/NMDAR blockade would suppress CPP acquisition via inhibiting LTP induction at glutamatergic inputs activated by contextual cues of the CPP box, while blocking potentiated NMDAR-mediated excitation at those inputs would interfere with CPP expression".*

This sentence has been removed.